# Determination of Acrylamide in Selected Foods from the Romanian Market

**DOI:** 10.3390/foods10092110

**Published:** 2021-09-06

**Authors:** Elena Narcisa Pogurschi, Corina Aurelia Zugravu, Ioan Nicolae Ranga, Svetlana Trifunschi, Melania Florina Munteanu, Dana Catalina Popa, Minodora Tudorache, Ioan Custura

**Affiliations:** 1Formative Sciences in Animal Breeding and Food Industry Department, Faculty of Animal Productions Engineering and Management, University of Agronomic Sciences and Veterinary Medicine, 57 Marasti Blvd, 011464 Bucharest, Romania; elena.pogurschi@usamv.ro; 2Department of Fundamental Sciences, Faculty of Nursing and Midwifery, University of Medicine and Pharmacy Carol Davila, 37 Dionisie Lupu Street, 020021 Bucharest, Romania; dr_corinazugravu@yahoo.com; 3Industrial Biotechnology Department, Faculty of Biotechnology, University of Agronomic Sciences and Veterinary Medicine, 57 Marasti Blvd, 011464 Bucharest, Romania; ionut_ranga@yahoo.com; 4Department of Pharmaceutical Sciences, Faculty of Pharmacy, Vasile Goldis Western University of Arad, 86th Liviu Rebreanu Street, 310048 Arad, Romania; trifunschi.svetlana@uvvg.ro (S.T.); munteanu.melania@uvvg.ro (M.F.M.); 5Production and Processing Technologies Department, Faculty of Animal Productions Engineering and Management, University of Agronomic Sciences and Veterinary Medicine, 57 Marasti Blvd, 011464 Bucharest, Romania; danasandulescu@yahoo.com (D.C.P.); johncustura2000@yahoo.com (I.C.)

**Keywords:** acrylamide, bread, exposure, Arabica ground roasted coffee, potato chips, pretzels

## Abstract

Several processed food products may contain toxic compounds considered risk factors for human health. Known for its possible carcinogenic effects, acrylamide is an organic compound periodically analyzed by the entities responsible for consumer safety. Knowing the acrylamide content of food offers the possibility of implementing corrective measures when needed, targeted at lowering its level. The aim of the paper was to screen for the presence of acrylamide in four products consumed almost daily in Romania and calculate acrylamide exposure by consuming one serving. Expressed in µg/kg coffee has the highest average acrylamide content (199), followed by potato chips (134), pretzels (120), and bread (14). Results regarding the acrylamide content in one serving showed the highest levels of acrylamide in pretzels (10.20 µg/serving), followed by potato chips (4.00 µg/serving), coffee (2.20 µg/cup), and bread (0.40 µg/slice). The calculation of the acrylamide content for one serving of the studied products will facilitate the following studies on the dietary acrylamide intake of the Romanian population, studies which, to our knowledge, have not been performed so far.

## 1. Introduction

Food safety is one of the priority areas in European policies. Beyond providing all the nutrients, food must be free of any contaminant that could lead, whether in a shorter or longer period of time, to the illness. The risk assessment of different contaminants is the responsibility of European Food Safety Authority European Food Safety Authority (EFSA), which draws up opinions based on the scientific evidence available at a given time, and the risk management is the responsibility of the European Commission, through expert groups, which implements legislative measures to protect the European consumer. Among the substances that have been evaluated and whose limitation is desired is acrylamide. This is an organic compound (CH_2_=CHC(O)NH_2_) that is found in various products that are not used as food, for example in plastics or paper. However, studies have shown that it can be generated in food, during their thermal preparation, by the well-known and often desired Maillard reaction (Figure 1), due to the reaction of free amino acids, predominantly asparagine, with the carbohydrates present in the product [1,2,3]. Of the six product categories formed, the health implications have premelanoidins and melanoidins. Premelanoidins have a stable and reactive structure, which through condensation and polymerization pass into melanoidin pigments. The melanoidin pigments are complex polymers responsible for the brown color in baked, roasted, or fried foods such as fried potato, potato chips, coffee and cereal-based products.

Recent studies have shown that humans ingest daily appreciable amounts of melanoidins, but their metabolism has not been studied enough. The adverse effect of melanoidins on the human large intestine microbiota has been demonstrated. However, no relevant evidence on human health has been established [4].

An already classic study conducted in 2002 in Sweden associated exposure to acrylamide with the risk of cancer, although some successive studies have tried to deny this effect [5] or gave mixed results [6]. Although many epidemiological studies have attempted to demonstrate the relationship between dietary acrylamide exposure and various types of cancer, there has been insufficient evidence to support this. However, in Denmark, a recent study highlighted the relationship between dietary acrylamide exposure and breast cancer [7]. Raw foods are free of acrylamide, while processed foods at temperatures above 120 °C or higher contain varying proportions of acrylamide depending on the processing temperature and its time as Dybing et al., reported in (2005) [8]. Variable amounts of acrylamide are found in breakfast cereals, French fries, bread, and bakery products, as well as roasted coffee and potato chips. Reported acrylamide concentrations vary widely from product to product (325 µg/kg corn chips, 71 µg/kg pretzels, 32 µg/kg toasted bread [9], country to country (325 µg/kg potato chips in China [10] to 4000 µg/kg potato chips in Sweden [11]), and brand to brand for the same product.

At this moment, the International Agency for Research on Cancer [12] has considered acrylamide as a “probable human carcinogen” and the US Environmental Protection Agency, as “likely to be carcinogenic to humans” [13]. In the European Union, particular attention has been paid to acrylamide, from assessing its level in food to developing methods to reduce its formation and, therefore, the quantity ingested [14]. Even if there is currently no maximum level for acrylamide in diet, based on the EFSA assessment in 2015 [5] it was considered necessary to draw up a regulation covering the subject. It is true that the existing data did not show a clear carcinogenic effect in humans, but in the calculation of the margins of exposure (MOE) it was concluded that there are suspicions of carcinogenic effect, based on evidence from data originating in experimental animal studies. As a consequence, a regulation has been elaborated, the commission Regulation 2017/2158 of 20 November 2017 establishing mitigation measures and benchmark levels for the reduction of the presence of acrylamide in food [15], its purpose being not the complete prohibition of acrylamide in food but to offer a plan of measures that can consistently reduce its level, without negative effects on the products on which they are applied. In the light of this regulation, we note the need for continuous assessment of the level of acrylamide in heated products that have been shown to possibly contain high levels and that are high contributors for human ingestion, both in order to follow the adequacy in relation to the benchmark levels, and to evaluate the possible effectiveness of the reduction measures applied. A diet free of substances with carcinogenic potentials, such as acrylamide, has been the subject of numerous studies, but, to our knowledge, no studies have been published regarding the acrylamide content of a serving of food. In this study, we evaluated the levels of acrylamide in four products with a high risk of exceeding the acrylamide benchmark collected from the Romanian market in the last half of 2019: Arabica ground roasted coffee, white bread, pretzels, and potato chips. The four products were chosen in the context in which [16] they were considered the main contributors of exposure to acrylamide in the diet of adults. The risk of exposure will be correlated with the number of servings consumed and their frequency in a future large-scale study.

## 2. Materials and Methods

Samples from the four food groups were obtained from different, randomly selected shops in Bucharest. The determination of acrylamide from the four food matrices was performed by LC-MS/MS, in accordance with the working methods described by [17].

### 2.1. Chemicals and Reagents

Acrylamide Standard D3 was acquired from LGC Standards, Wesel, Germany. Acetone GC 99% purity was acquired from Carlo-Erba, Milano, Italy. Methanol (UHPLC Grade) was acquired from Carlo-Erba. Salts for QueEChERS: MgSO_4_ 99.5% anhydrous (4 g) + NaCl 99.8% p.a (0.5 g) were purchased from Celera-Chemie, Sofia, Bulgaria. Disodium citrate was acquired from Celera-Chemie.

### 2.2. Calibration Solutions Preparation

The calibration solutions were prepared according to EN 16618:2015. The two stock solutions were prepared in advance, namely the stock standard solution and the stock internal standard solution by dissolving 100 ± 0.05 mg of acrylamide standard and deuterated acrylamide (AA-D_3_) in 100 mL water. Next, aliquots of standard solutions were diluted with water to obtain acrylamide calibration solutions in the range of 2–50 µg/L. Each level of calibration contained the same concentration of internal standard as the prepared samples.

### 2.3. Samples Preparation

Samples Preparation from bakery products and potato products

2 g of the crushed and homogenized test sample were weighed into the centrifuge tube. 400 µL internal standard (deuterated acrylamide) of 1 mg/L concentration and 40 mL water was added. The centrifuge tube was shaken vigorously, vortexed for one min, and then stirred mechanically for one hour. After homogenization, the sample was centrifuged for 20 min at 10 °C, and then 10 mL of aqueous extract was transferred to a clean test tube.

Samples Preparation from ground roasted coffee

2 g of Arabica ground coffee were placed in a centrifuge tube over which 400 µL internal standard (deuterated acrylamide) of concentration 1 mg/L, 5 mL n-Hexane, and 40 mL water was added. The tube was stirred vigorously, vortexed for 1 min, then stirred with the mechanical stirrer for one hour. It was centrifuged for 20 min at 10 °C, the organic phase was removed and 10 mL of aqueous extract was transferred to a clean test tube.

### 2.4. Method Validation

Quantification for acrylamide content from the obtained extracts was performed by liquid chromatography tandem mass spectrometry (LC-MS/MS).

The extraction of acrylamide from food samples was performed with ultrapure water by liquid–liquid (LL) and/or liquid–solid (LS) partition. After homogenization and centrifugation, the aliquot extract was purified by LL or LS partition. The purified aliquot extract was analyzed as such by injection into LC-MS. Chromatographic separation was performed on the Luna C18 column (150 × 3 mm inner diameter, 3 um particle size), maintained at 60 °C. Mobile phases consisted of an aqueous phase (water with 0.1% acetic acid, *v/v*) and an organic phase (methanol with 0.1% acetic acid, *v/v*), at a flow rate of 0.25 mL/min using a gradient elution program. The method is compatible with the European Committee for Standardization standard methods, EN 16618:2015 (Food analysis—Determination of acrylamide in food by liquid chromatography tandem mass spectrometry (LC-MS/MS)). Table 1 provides the results of method validation.

To ensure the capability of the LC-MS/MS method for the conducted study, linearity, limit of detection (LOD) and limit of quantification (LOQ), specificity, precision, and accuracy were selected as defining parameters to evaluate the performance characteristics.

Acrylamide determinations were performed for the following four products that are considered everyday products [18]: Robusta ground roasted coffee, white bread, pretzels, and potato chips. For each product, 10 samples were taken from different brands (10 samples/brand). The most common brands were analyzed. The total number of analyzed samples was 400 (4 products × 10 brands × 10 repetitions), the average acrylamide content was determined for one serving of each product. Results were expressed as µg/kg wet mass. The following sections describe the details of the validation method.

### 2.5. Limit of Detection (LOD), Limit of Quantification (LOQ)

The limit of quantification/determination (LOQ) was 12 µg/kg and is specified in tables of analysis (Table 2, Table 3, Table 4 and Table 5), the detection limit (LOD) was 3.6 µg/kg. The limit of quantification was calculated as 10 × s, where s = the standard deviation of a sample with content close to the first point of the calibration curve (10 replicates of the analyzed sample), in absolute value in solution (2.5 µg/L). The value of 12 µg/kg is the maximum value obtained for the types of samples analyzed, obtained in the validation process of the analysis method.

### 2.6. Linearity and Calibration Curve

For the quantitative determination of acrylamide from the targeted matrixes, the instrument was calibrated using five standard solutions of acrylamide within the range 2–50 µg/L and an internal standard of 13C_3_-labelled acrylamide standard. The coefficient of determination (R^2^) value was 0.9999, which indicates excellent linearity (y = 0.02243x − 0.0008716).

### 2.7. Specificity

The response of the analyte in the targeted matrixes containing the analyte and all potential sample components is compared with the response of a solution containing only the analyte. Retention time did not show any significant change between different matrixes with an average retention time of 3.55 ± 0.04 min.

### 2.8. Precision

Repeatability was determined with the method operating over a short time interval under the same conditions. For this purpose, 10 repeated injections were performed from each of the samples of interest. Repeatability values were found in the range of 2.3–4% for all targeted samples. Intermediate precision was performed on different days and by different analysts. Similarly, 10 repeated injections from each sample were performed and the range was found to be <5%.

### 2.9. Accuracy

The accuracy study was performed by standard additions by spiking analyte in the targeted sample. Spiked samples have been prepared in triplicate at three levels over a range of 50–150% of the target concentration. The percentage of recovery was calculated and values were found in the range of 80–120% for all sample types.

### 2.10. Acrylamide Analyses by LC-MS/MS

The LC-MS/MS (UHPLC:Ultimate 3000, MS: Endura) equipment, produced by Thermo Scientific runs the TraceFinder EFS Software (Waltham, MA, USA). The samples were separated on a Luna C18 column (150 × 3 mm, 3 µm, 100 A; Phenomenex, Torrance, CA, USA) at 25 °C. The mobile phase was composed of solvent (A) 0.001% (*v/v*) aqueous formic acid solution and solvent (B) methanol was used with 0.1% (*v/v*) formic acid. The following gradient program was applied from 0 to 10 min 10–0% B, from 10 to 15 min 0–10% B. The injection volume was 3 μL and the flow was 0.200 mL/min. The optimization of the compounds was performed by injecting directly into the MS, the standards in order to establish the transitions (ionization products), the collision energy of each transition, and the capillary tension. The identification of the components in the sample was based on the retention time, electronic transitions, and collision energies for LC-MS/MS. The determination was made by using an integration system, performing the automatic marking of the retention times and the integration of the chromatographic peaks, directly displaying their area and the concentration of the extract from the 50 mL (CA) extract.

## 3. Results and Discussion

### 3.1. Acrylamide Content in Arabica Ground Roasted Coffee and the Exposure through One Cup of Coffee

In order to be able to make valid comparisons between the 10 brands of coffee analyzed, it was necessary to grind the coffee to a uniform fine grind, respectively 0.60 mm. The amount of acrylamide detected in 1 kg of Arabica ground roasted coffee per each brand analyzed is presented in Table 2.

The lowest average level of acrylamide was 85.6 µg/kg and the highest average level of acrylamide, found in Brand 3 was 374 µg/kg, which means it is 6.5% less compared to the benchmark level (400 µg/kg) specified by the legislation in force [19]. The average level of acrylamide in all the analyzed samples was 195.96 µg/kg, which means it is 51% less compared to the benchmark level allowed by European legislation. Exposure to acrylamide by consuming a cup of brewed coffee (300 mL) was calculated using the conversion factor (26 ± 3) indicated by Andrzejewski et al. (2004) [20]. On average, by consuming a 300 mL brewed coffee cup, the exposure to acrylamide is 2.20 µg/300 mL. The conversion factor proposed by Andrzejewski et al. (2004) [20] is similar to that proposed by Abt et al. (2019) [21].

In the EFSA Comprehensive database, we could not find figures regarding coffee consumption in Romania. However, considering that the average Romanian adult drinks 1 cup of coffee/day (as unpublished data of food and beverage consumption show), the exposure to acrylamide through the consumption of prepared coffee is 2.20 µg. Exposure to acrylamide by coffee is lower compared to the exposure recorded in Spain of about 2.64 µg/person/day but at a consumption rate of 1.64 cups of coffee/person/day [22].

According to the EFSA Scientific Committee [23] a 70 kg adult is the reference for calculating the exposure per kg body weight (bw). In this context, the exposure by consuming a cup of coffee/day is on average 0.031 µg/kg bw/day. This result is comparable to acrylamide exposure through coffee consumption reported by Mesias et al. (2016) [22] (0.037 µg/kg bw/day). However, higher values were registered in France (0.168 µg/kg bw/day), Sweden (0.171 µg/kg bw/day), and Denmark (0.106 µg/kg bw/day) [24].

Coffee might be, in the EFSA’s opinion, a contributor to acrylamide intake. However, the highest levels of acrylamide were not found in roasted coffee, but in ‘Coffee substitutes (dry)’ (average medium bound (MB) levels of 1499 µg/kg) and ‘Coffee (dry)’ (average medium bound (MB) levels of 522 µg/kg). However, due to dilution effects, lower levels are expected in ‘Coffee beverages’ and ‘Coffee substitute beverages’ as consumed by the population [25]. In light of these figures, substituting coffee with coffee substitutes such as roasted chicory root or various roasted cereals is not a solution, since the EFSA draws the conclusion that they might have greater levels of acrylamide, chicory in particular.

### 3.2. Acrylamide Content in White Bread and the Exposure through One Serving

The acrylamide content of the 10 white bread brands analyzed and the average acrylamide content of the white bread produced and marketed in Romania is presented in Table 3. The lowest average level of acrylamide was 9.80 µg/kg and the highest average level of acrylamide, found in Brand 6 was 17.30 µg/kg, which means it is 34.6% less compared to the benchmark level (50 µg/kg) specified by the legislation in force [19]. The average level of acrylamide in all the analyzed samples was 13.63 µg/kg, which means it is 27.26% less compared to the benchmark level allowed by European legislation. Compared to Croatia, where the market of bread and bakery products has a main ingredient of wheat flour, it was found that the bread produced in Romania had a much lower content (73%) of acrylamide (13.63 µg/kg compared to 51 µg/kg) [26].

There are no standardized serving sizes in Europe [27], thus creating problems both for consumers and for producers, obliged to respect regulation 1169/2011. A need for standardization has been underlined, but it has not been achieved yet. Producers of bread usually use one slice of toast as a portion size, having approximately 28–30 g. The serving size of bread is influenced by the type of cereal used, the processing aids, and last but not least, the thickness of the slice of bread. For white bread, the type of bread for which the acrylamide content was determined, and for an average serving of about 30 g the acrylamide content was 0.40 µg/serving.

Figures from the EFSA Comprehensive database show an average consumption per adult of 176.88 ± 93 g/day of grains and grain-based products, based on an intake study from 2012. Intake of the most popular bread (white bread) was 94.16 ± 14 g/day per adult [28]. More recently, according to Eurostat data, in Romania, the population consumes approx. 260 g of bread per day, which means about nine servings of bread daily. Romania is considered one of the biggest bread consumers in Europe. This consumption is equivalent to an intake of about 3.6 µg acrylamide/person/day. The acrylamide intake through the consumption of one serving of bread is very low (0.005 µg/kg bw/day) but the total exposure depends on the number of servings consumed/day or on consumption habits. The daily exposure to acrylamide through the consumption of bread calculated at a consumption rate of nine servings per day is 0.045 µg/kg bw/day.

Compared to the average consumption of bread in Europe (181 g/day), in Romania, a 43.6% higher consumption rate can be observed. It is most likely this high consumption rate is due to the low price of bread compared to other staple foods.

Even though the level of acrylamide in bread is moderate, it can contribute consistently to the intake of acrylamide, due to the constant and high intake of bread.

### 3.3. Acrylamide Content in Pretzesl and the Exposure through One Serving

In Romania, active people prefer to consume 1–2 pretzels in the morning instead of having a healthy breakfast. Similar to the Austrian pretzel, the pretzel in Romania weighs on average cc. 85 g and is preferred in the crunchy version and much better baked. The average acrylamide content in pretzels from the most popular brands analyzed and the acrylamide content/serving (1 pretzel) is presented in Table 4.

The pretzel market in Romania is huge and statistics show that annually it has a value of approximately 160 million euros. According to the latest statistical data, one out of six Romanians consume a pretzel daily, which is why the most popular pretzel brands and their acrylamide content were analyzed. The lowest acrylamide content in pretzels was observed in Brand 2, respectively 80.50 µg/kg, which is equivalent to 10.2 µg/serving (1 piece). The maximum acrylamide content was found in Brand 3 −159.00 µg/kg, the value corresponding to 13.51 µg/serving (1 piece). The acrylamide content from pretzels produced in Romania is in line with the values reported for the same product in Latvian studies (39–588 µg/kg) as was reported by Pugajeva et al. (2014) [29]. The average level of acrylamide in all analyzed samples was 120.00 µg/kg which means it is 28 µg/kg more than the values reported for the same product in Lesser Poland by Cieślik et al. (2020) [30]. The average pretzel concentration was also higher than that obtained by Normadin et al. (2013) [9] in Canada (71 µg/kg).

The acrylamide intake through consumption of one pretzel is on average 0.145 µg/kg bw/day. Pretzels, as well as potato chips, are known to be the favorite snacks of children and teenagers. It is easy to intuit that they do not stop at consuming only a serving of these snacks. These two age categories are expected to have a much higher exposure to acrylamide through diet. The Joint FAO/WHO Expert Committee on Food Additives in 2005 noted that children may have intakes of acrylamide around two or three times higher than adults when it is expressed in kg bw [31].

### 3.4. Acrylamide Content in Potato Chips and the Exposure through One Serving

Table 5 provides the data on the acrylamide content of the analyzed potato chip samples. The lowest acrylamide content in potato chips was observed in Brand 10, respectively 82.00 µg/kg, which is equivalent to 2.46 µg/serving (30 g).

The maximum acrylamide content was found in Brand 7 −266.00 µg/kg, a value corresponding to 7.98 µg/serving (30 g). Our results are consistent with those reported by Arisseto et al. (2009) [32], who also observed a very large variation of acrylamide content in potato chips from different brands (from 591 to 1999 µg/kg). Wide variations in acrylamide content for the same product manufactured by different brands or even within the same brand can be attributed to manufacturing conditions (baking temperature, frying time), factors that greatly influence the formation of this toxic compound in food [33]. The average level of acrylamide in all the analyzed samples was 133.60 µg/kg which means it is 55.46% less compared to the benchmark level allowed by European legislation (300 µg/kg). In 2015, Oroian et al. (2015) [34] reported 1782 µg/kg acrylamide concentrations for 50 potato chip samples collected from the Romanian market. The decrease of acrylamide concentration in potato chips has been more than 10% in the last five years, which is why we can say that not only the processing time and cooking method contributed to this fact but also the manufacturers’ practices as highlighted [19]. The wide variations of the acrylamide levels, not only between the four analyzed products but also within each food category, are similar to other reported studies [35,36].

The acrylamide intake through consumption of one serving of potato chips is 0.057 µg/kg bw/day. Daily exposure to acrylamide, however, depends on the number of servings consumed, which in the case of potato chips is hard to believe is just one.

For a good representation of the acrylamide content from analyzed products that facilitates the consumer in choosing those with a minimum acrylamide content per serving, the average acrylamide content is represented graphically in Figure 2.

## 4. Conclusions

In our study, four products were analyzed in terms of acrylamide content. Four categories of products were selected because they represent the main sources of acrylamide in the daily diet in accordance with the intake studies reported by EFSA, 2011. In order to facilitate further studies on the exposure to acrylamide of the Romanian population through diet, the average acrylamide content of one serving of each product was calculated. Expressed in µg/kg, coffee has the highest average acrylamide content (195.96), followed by potato chips (133.60), pretzels (120.00), and bread (13.63). Results regarding the acrylamide content in one serving showed the highest levels of acrylamide in pretzels (10.20 µg/serving), followed by potato chips (4.00 µg/serving), coffee (2.20 µg/cup), and bread (0.40 µg/slice). The four groups of foods analyzed are widely consumed foods, at least one of them being present in any European citizen’s daily diet, thus contributing to the acrylamide intake. We underline the importance of the periodic monitoring of acrylamide levels in such foods, in order to take lowering measures when levels rise. European consumers have to have lower exposure to the contaminant, in order to reduce the risk of further health consequences.

## Figures and Tables

**Figure 1 foods-10-02110-f001:**
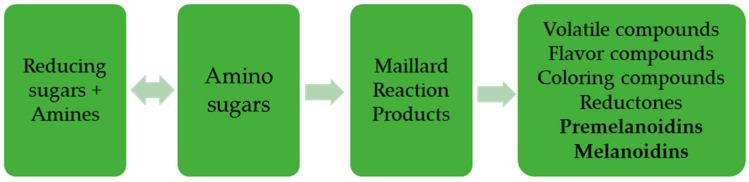
Maillard Reaction.

**Figure 2 foods-10-02110-f002:**
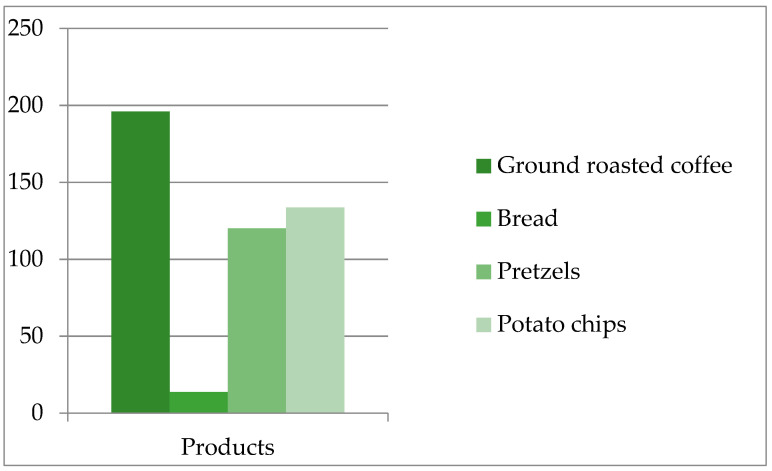
Acrylamide content of ground roasted coffee, bread, pretzels, potato chips (µg/kg).

**Table 1 foods-10-02110-t001:** Method validation.

Type	Linearity		Precision		Accuracy		Limits	
	Range(µg/L)	R^2^	Intraday (%)	Interday(%)	Recovery(%)	RSD(%)	LOD(µg/kg)	LOQ(µg/kg)
Standard Solution	2–50	0.9999	0.8	1.4	98.6	1.1	3.6	12
Coffee			2.6	3.6	102.6	1.8	3.6	12
Bread			2.3	4.1	96.9	3.6	3.6	12
Pretzel			4	4.9	116.5	2.8	3.6	12
Potato chips			3.1	3.7	108.2	3.1	3.6	12

RSD–relative standard deviation; LOD–limit of detection; LOQ–limit of quantification.

**Table 2 foods-10-02110-t002:** The acrylamide level in Arabica ground roasted coffee (µg/kg).

Product	Acrylamide (µg/kg)			Determination Limit (µg/kg)	Measurement Uncertainty ± Uext (µg/kg)	Benchmark Level (µg/kg)
	Minimum	Mean	Maximum			
Brand 1	69.5	85.6	101.7			
Brand 2	118.5	146	173.5			
Brand 3	321	374	427			
Brand 4	89	197	305			
Brand 5	112	205	298			
Brand 6	164	192	220	12	±29.8	400
Brand 7	94	126	158			
Brand 8	241	319	397			
Brand 9	112	150	188			
Brand 10	143	165	187			
		195.96				
µg/300 mL of brewed coffee		2.20				

**Table 3 foods-10-02110-t003:** The acrylamide level in white bread (µg/kg).

Product	Acrylamide (µg/kg)			Determination Limit (µg/kg)	Measurement Uncertainty ± Uext (µg/kg)	Benchmark Level (µg/kg)
	Minimum	Mean	Maximum			
Brand 1	13.00	14.00	15.00			
Brand 2	14.80	15.15	15.50			
Brand 3	11.00	14.00	17.00			
Brand 4	9.80	12.00	14.20			
Brand 5	11.20	13.10	15.00			
Brand 6	15.80	16.55	17.30	12	±6	50
Brand 7	10.80	11.35	11.90			
Brand 8	12.00	14.00	16.00			
Brand 9	10.10	11.45	12.80			
Brand 10	13.00	14.70	16.40			
		13.63				
µg/serving (30 g bread/slice of bread)		0.40				

**Table 4 foods-10-02110-t004:** The acrylamide level in pretzels (µg/kg).

Product	Acrylamide (µg/kg)			Determination Limit (µg/kg)	Measurement Uncertainty ± Uext (µg/kg)	Benchmark Level (µg/kg)
	Minimum	Mean	Maximum			
Brand 1	95.00	108.00	121.00			
Brand 2	80.50	96.00	111.50			
Brand 3	175.00	143.00	159.00			
Brand 4	126.00	128.50	131.00			
Brand 5	115.00	118.00	121.00			
Brand 6	84.0	99.00	114.00	12	±45.6	400
Brand 7	104.00	120.00	136.00			
Brand 8	120.00	134.00	148.00			
Brand 9	100.00	110.00	120.00			
Brand 10	140.00	143.50	147.00			
		120.00				
µg/serving (85 g^−1^ piece)		10.20				

**Table 5 foods-10-02110-t005:** The acrylamide level in potato chips (µg/kg).

Product	Acrylamide (µg/kg)			Determination Limit (µg/kg)	Measurement Uncertainty ± Uext (µg/kg)	Benchmark Level (µg/kg)
	Minimum	Mean	Maximum			
Brand 1	131.00	136.00	141.00			
Brand 2	106.00	119.00	132.00			
Brand 3	84.00	94.00	104.00			
Brand 4	104.00	116.00	128.00			
Brand 5	119.00	130.00	141.00			
Brand 6	168.00	180.00	192.00	12	±37.8	300
Brand 7	184.00	225.00	266.00			
Brand 8	101.00	110.00	119.00			
Brand 9	118.00	124.00	130.00			
Brand 10	82.00	102.00	122.00			
		133.60				
µg/serving (30 g)		4.00				

## Data Availability

Not applicable.

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
