# Peer review of "Determination of Acrylamide in Selected Foods from the Romanian Market"

_foods, 2021, doi:10.3390/foods10092110_

Round 1

Reviewer 1 Report

Determination of acrylamide in food has high importance for the consumers and for practice, as well. Therefore, the topic of manuscript can be considered as relevant and interesting. The manuscript is generall well structured. Authors have revised the manuscript thoroughly according to reviewers’ comments and suggestions. After the revision, the overall scientific quality of manuscript has been improved significantly. I accept all modifications and recommend manuscript foods-1328889 for publishing.

Author Response

We are grateful for all the efforts made during the revision process of our manuscript, and consider that based on your recommendations and suggestions the quality of the article was much improved. 

Reviewer 2 Report

I have several comments and suggestions to your manuscript.

First of all I suggest to change a title a little bit, f.ex. by removing “Expose to…in Romania”, because it is slightly liming an interest of readers. Maybe focus on the content of the acrylamide in those products.

According to the Abstract: I would suggest to extend it by adding some basic information about acrylamide, what is it and why it is important to evaluate its content in food etc.

I suggest to improve the Figure 1 to make it more “nicely looking” as you decided to make the graph.

Line 18-19 and line 161-162 – provide some data from the literature proofing that those products are among daily consumed products. I think it would be interesting to report the consumption of those products in a wider group f.ex. in European Union countries, not to limit the exposure data only to Romania.

Line 70: remove an extra dot at the end of a sentence

Line 76: “maximum recommended levels of acrylamide” sounds misleading, replace with maximum reside level or maximum amount. Additionally, in the Results and Discussion section Line 228 the Authors mentioned the MRL of 400ug/kg and later on, another MRL for another products, so the data are inconsistent.

Materials and Methods section: I think you should provide more information about the analysed products, at least a country of origin and the main raw materials used in production, f.ex. is it Arabica coffee or a mixture with Robusta? Also bread is a wide category, please specify it. It would be also important to provide the year in which the products were taken from the market.

Line 164: It is not clear why you analysed 400 samples? 4 products * 10 brands * 10 repetitions?

As a results are expressed per wet mass in a material it would be good to provide a water content for a better characteristic of the analysed products.

Line 271: what do the Authors mean by “Coffee substitutes” – characterise it shortly.

Line 289-290: please provide a citation, because as far as I know 50g of bread is considered as a serving.

Line 321: I do not see a link why the Authors suddenly mention about a meat in this paper. In my opinion it is to much out of the context.

Line 325: What the Authors mean by “active consumer”?

Figure 2: I suggest to make a standard 2D graph because in this form it is much more difficult to read the amount of acrylamide and to compare the products. Is it per kg of wet or dry basis?

Author Response

we are grateful for all the efforts made during the revision process of our manuscript, and consider that based on your recommendations and suggestions the quality of the article was much improved. We appreciate the overall positive feed-back received from you, and the general remarks on the quality of our work. Please find our point-by-point answers to your comments and remarks bellow:

  1. First of all I suggest to change a title a little bit, f.ex. by removing “Expose to…in Romania”, because it is slightly liming an interest of readers. Maybe focus on the content of the acrylamide in those products

Changed as suggested, please see Line 1 (Levels of Acrylamide in daily foods: are they a reason for concern?!)

  1. According to the Abstract: I would suggest to extend it by adding some basic information about acrylamide, what is it and why it is important to evaluate its content in food etc.

Inserted as suggested, please see Lines 15-20

Several processed plant products contain, in addition to the constituents that bring benefits to the body, toxic compounds considered risk factors for human health.

Known for its possible carcinogenic effects, acrylamide is an organic compound periodically analyzed by the entities responsible for consumer safety that prepares reports on its presence in various foods. Knowing the acrylamide content of food offers the possibility of creating complex connections between ingested foodstuff and health.

  1. I suggest to improve the Figure 1 to make it more “nicely looking” as you decided to make the graph.

Improved as suggested, please see Lines 50-53

  1. Line 18-19 and line 161-162provide some data from the literature proofing that those products are among daily consumed products. I think it would be interesting to report the consumption of those products in a wider group f.ex. in European Union countries, not to limit the exposure data only to Romania.

Inserted as suggested, please see Line  154

Acrylamide has been found in a wide variety of cooked foods, including those prepared industrially, in catering and at home. It is found in staple foods such as bread and potatoes as well as in other everyday products such as crisps, biscuits and coffee. (https://www.fooddrinkeurope.eu/resources/publications)

In the context of the approached subject, we compared the acrylamide content of heated everday products from different countries.

For coffee: Spain, France, Sweden, and Denmark. Lines: 224-229; 255-257

For white bread: Croatia Line 276;

For pretzel: Poland, Canada Lines 361; 363

For potato chips: Brazilia, Canada Lines 397; 410

The quantities consumed were strictly described for the calculation of the acrylamide content per serving, which, to our knowledge, other authors have not reported.

  1. Line 70:remove an extra dot at the end of a sentence

Removed as suggested.

  1. Line 76: “maximum recommended levels of acrylamide” sounds misleading, replace with maximum reside level or maximum amount. Additionally, in the Results and Discussion section Line 228the Authors mentioned the MRL of 400ug/kg and later on, another MRL for another products, so the data are inconsistent.

Replaced as suggested, please see Line 77

Maximum recommended levels of acrylamide was replaced by maximum level for  acrylamide.

We removed the misunderstandings, see Lines 83, 216,218, 271,273, 403.

In the Results and Discussion MRL was replaced by ″Benchmark level″ , the term being defined and specified in the Reg UE 2158/2017. Benchmark levels for the presence of acrylamide in foodstuffs, are set for each product category, which is why different values ​​are mentioned from one product to another (400ug/kg roasted coffee, 50ug/kg bread, etc.).

  1. Materials and Methods section:I think you should provide more information about the analysed products, at least a country of origin and the main raw materials used in production, f.ex. is it Arabica coffee or a mixture with Robusta? Also bread is a wide category, please specify it. It would be also important to provide the year in which the products were taken from the market.

Inserted as suggested, please see Lines 95,96, 208,212,267,268,284,308.

At your suggestion, we specified the types of products, respectively Robusta coffee and white bread. This specification has been made throughout the text where these products appear.We also specified the year of products collection (collected from the Romanian market in the last half of 2019). Inserted as suggested, please see Line 95.

  1. Line 164: It is not clear why you analysed 400 samples? 4 products * 10 brands * 10 repetitions?

Changed and inserted as suggested, see Line 157-158

We have added your suggestion and it is much clearer now the total number of samples analyzed (400).

  1. As results are expressed per wet mass in a material it would be good to provide a water content for a better characteristic of the analysed products.

Thank you for drawing our attention to this very useful remark.

In order to compare with the legislation in force and with the data reported by other authors, the acrylamide content was reported per kg product not per kg of dry product.

  1. Line 271: what do the Authors mean by “Coffee substitutes” – characterise it shortly.

like roasted chicory rootor various roasted cereals;

Inserted as suggested, please see Line 264.

  1. Line 289-290:please provide a citation, because as far as I know 50g of bread is considered as a serving.

Inserted as suggested, please see Lines 277-282

According to the United States Department of Agricuture a single serving of bread is considered one slice which is equivalent to 1 ounce (28,3495 g). The serving size of bread is influenced by the type of cereal used, the auxiliary ingredients used and last but not least the thickerness of the slice of bread. For white bread, type of bread for which acrylamide was determined and for an average serving of about 30 g the acrylamide intake was 0.40µg/serving. 

  1. Line 321:I do not see a link why the Authors suddenly mention about a meat in this paper. In my opinion it is to much out of the context.

Removed as suggested.

It was wanted to highlight the consumption of basic nutrients of the rural population.

Considering that it was not necessary, we eliminated this comparison.

  1. Line 325:What the Authors mean by “active consumer”?

Replaced as follow, please see Line 323

The term ″active consumer″ was replaced by ″the active person″, thus avoiding confusion with the term used in the economics field.

  1. Figure 2:I suggest to make a standard 2D graph because in this form it is much more difficult to read the amount of acrylamide and to compare the products. Is it per kg of wet or dry basis?

Rearranged as suggested, please see Lines 418-420 (Figure 2)

In order to compare with the legislation in force and with the data reported by other authors, the acrylamide content was reported per kg product not per kg of dry product.

We hope that we addressed the questions and remarks raised in a proper manner, and that you will find our manuscript to be suited for publication in the current form. Again, we wish to thank you for your efforts during the manuscript revision, and for the valuable suggestions and comments.

Round 2

Reviewer 2 Report

Thank you for the revision of the manuscript. I think it was significantly improved. The new title sounds much more interesting.

I wanted to read a new reference, but unfortunately, it is not accessible please check the link again.(FoodDrinkEurope.Acrylamide toolbox 2019, Available from https://www.fooddrinkeurope.eu/uploads/publications_documents/FoodDrinkEurop e_Acrylamide_Toolbox_2019.pd, ac-521 cessed 15.012.2020.) I think there is also a typo in the date. Additionally, it would be recommended to provide a link that is accessible at least at the date of writing, not from a previous year.

"Acrylamide has been found in a wide variety of cooked foods, including those prepared industrially, in catering, and at home. It is found in staple foods such as bread and potatoes as well as in other everyday products such as crisps, biscuits, and coffee. (https://www.fooddrinkeurope.eu/resources/publications)" I still have some doubts whether pretzels are within the daily foods. Anyway "in the context of the approached subject, we compared the acrylamide content of heated everyday products from different countries." it makes sense, however, it is still not clear whether the Authors focused on the exposure in Romania or different countries?

According to the serving size of the bread. In general, if you study apply to Europe it is better to provide a serving size in Europe, not in the USA. "According to the United States Department of Agriculture a single serving of bread ..."

Please provide some adjustments and improvements.

Author Response

Esteemed reviewer,

please find enclosed to our submission the manuscript entitled ″ Levels of Acrylamide in daily foods: are they a reason for concern?!″, authors Elena Narcisa Pogurschi, Corina Aurelia Zugravu, Ranga IonuČ›, Svetlana Trifunschi, Melania Florina Munteanu, Popa Dana Catalina, Tudorache Minodora and Custura Ioan, that we have resubmitted for publication as a ″Original article″ to the FOODS - sections Food Quality and Safety. We appreciate the overall positive feed-back received from you, and the general remarks on the quality of our work.

As you suggested we made the changes and we put these in yellow colour to be easily remarked as well as we cited the line number and exact changes. All the responses are following:

Point 1. I wanted to read a new reference, but unfortunately, it is not accessible please check the link again.(FoodDrinkEurope.Acrylamide toolbox 2019, Available from https://www.fooddrinkeurope.eu/uploads/publications_documents/FoodDrinkEurop e_Acrylamide_Toolbox_2019.pd, ac-521 cessed 15.012.2020.) I think there is also a typo in the date. Additionally, it would be recommended to provide a link that is accessible at least at the date of writing, not from a previous year.

Response 1

https://www.fooddrinkeurope.eu/wpcontent/uploads/2021/05/FoodDrinkEurope_Arylamide_Toolbox_2019.pdf accessed 23.08.2021.

Please see the Lines: 487-488

Point 2. "Acrylamide has been found in a wide variety of cooked foods, including those prepared industrially, in catering, and at home. It is found in staple foods such as bread and potatoes as well as in other everyday products such as crisps, biscuits, and coffee. (https://www.fooddrinkeurope.eu/resources/publications)" I still have some doubts whether pretzels are within the daily foods. Anyway "in the context of the approached subject, we compared the acrylamide content of heated everyday products from different countries." it makes sense, however, it is still not clear whether the Authors focused on the exposure in Romania or different countries?

Response 2

A comparison of acrylamide exposure from different countries was made with the calculated exposure by analyzing the products under study.

Point 3. According to the serving size of the bread. In general, if you study apply to Europe it is better to provide a serving size in Europe, not in the USA. "According to the United States Department of Agriculture a single serving of bread ..."

Response 3

There is no standardized serving sizes in Europe [27], thus creating problems both for consumers, and for producers, obliged to respect regulation 1169/2011. A need for standardization has been underlined, but it has not been achieved yet. Producers of bread usually use as portion size a slice of toast, having approximately 28-30g. 

Please see the Lines: 276-279 

We believe that the quality of the manuscript was much improved based on your recommendations. We trust that you will find our manuscript to meet the requirements for publication in the journal. Again, we wish to thank you for your efforts during the manuscript revision, and for the valuable suggestions and comments.

This manuscript is a resubmission of an earlier submission. The following is a list of the peer review reports and author responses from that submission.

Round 1

Reviewer 1 Report

The manuscript is based on the determination of acrylamide in 4 products. The results are not of scientific soundness but contribute to the literature. Some issues in the manuscript should be revised.

Introduction: authors should provide more specific information about the toxicity of acrylamide. In addition, some reactions (as for example, Maillard reaction) should be added to improve the quality of the introduction section. Acrylamide structure should be included too.

Material&Methods: Figure 1 is not acceptable. It appears as a print screen from the PC. Please, replace that figure. Figure 2 has not enough quality.

Results&Discussion: An exposure assessment in children is recommended (overall that children consumes bread, pretzel and chips, not coffee).

Author Response

Dear Reviewer 1,

Thank you very much for your efforts during the revision process of our manuscript and the overall positive feedback.

Point 1: Introduction: authors should provide more specific information about the toxicity of acrylamide. In addition, some reactions (as for example, Maillard reaction) should be added to improve the quality of the introduction section. Acrylamide structure should be included too.

Response 1: More information about the toxicity of acrylamide was introduced in the Introduction section; please see lines (49-52); The general scheme of Maillard reaction also was introduced in this section, please see lines (39-44); Acrylamide structure was included, please see the line L34.

Point 2: Material & Methods: Figure 1 is not acceptable. It appears as a print screen from the PC. Please, replace that figure. Figure 2 has not enough quality.

Response 2: Figure 1 has been excluded from the text as you recommended. Regarding the figure 2, we have excluded the chromatograms and we included a graphic representation, as Reviewer 2 suggested. Please see the lines (374-375).

Point 3: Results & Discussion: An exposure assessment in children is recommended (overall that children consume bread, pretzel and chips, not coffee).

More information about exposure assessment in children was introduced in the Results and discussion section, please see lines (317-322).

We believe that the quality of the manuscript was much improved based on your recommendations. We trust that you will find our manuscript to meet the requirements for publication in the journal.

Reviewer 2 Report

The study of acrylamide determination in 4 food products in Romania is well executed and presented but I do not see much novelty. I would suggest to reveal the coffee brands for anyone interested.

Author Response

Dear Reviewer 2,

Thank you very much for your efforts during the revision process of our manuscript and the overall positive feedback. Please find our point-by-point answers to your remarks:

Point 1: The study of acrylamide determination in 4 food products in Romania is well executed and presented but I do not see much novelty. I would suggest revealing the coffee brands for anyone interested.

Regarding your suggestion to reveal coffee brands, we had consulted with the Legal Office of our university and they advised against the publications of commercial name of the brands, given the current legislation the intellectual property of companies and their associated brands. Therefore, we have to keep the commercial names anonymized.

Regarding the novelty of our study, we tried to improve the implications of our findings against the current literature, given that EFSA requests studies on the toxic components of the diet for the elaboration of new directives for the producers.

We believe that the quality of the manuscript was much improved based on your recommendations. We trust that you will find our manuscript to meet the requirements for publication in the journal.

Reviewer 3 Report

Investigation of the acrylamide concentration in food is an important topic, which can provide interesting information for the readers and consumers, as well. But the Introduction section is not complete: the average acrylamide concentration of different food, technological parameters affecting the acrylamide concentration are missing; the relevance of the selected food-ground roasted coffee, bread, pretzels and potato chips - is not underlined (daily consumption er capita), etc.

Materials and methods are adequate to the main aims of research and novel, these are given clearly.

In the manuscript, the acrylamide concentration of different food is discussed. The results can be considered valuable for the consumers.

Comments, suggestions:

I suggest the authors to give clearly the novelties of the study.

I suggest the authors to give information in Introduction section about the average acrylamide concentration in different food

Why is needed to give the calibration curve in the Manuscript (Figure 1)?

I suggest the authors to consider to present data in Figure to improve the visibility of experimental results.

I suggest the authors to give clear recommendations for the industry to decrease the acrylamide concentration of foods, and for consumer show to choose food to decrease the intake of acrylamide, respectively.

Author Response

Dear Reviewer 3,

Thank you very much for your efforts during the revision process of our manuscript and the overall positive feedback. Please find our point-by-point answers to your remarks:

Point 1: I suggest the authors to give clearly the novelties of the study.

Regarding the novelty of our study, we made efforts as to improve and highlight the implications of our findings, give that there is a lack of data and EFSA requests studies on the toxic components of the diet for the elaboration of new directives for the producers.

 Point 2: I suggest the authors to give information in Introduction section about the average acrylamide concentration in different food.

The average acrylamide concentration in different foods was introduced in introduction section, please see the lines 59-66.

Point 3: Why is needed to give the calibration curve in the Manuscript (Figure 1)?

Given your comment and the recommendation of Reviewer 1, we decided to exclude figure 1 from the revised manuscript.

Point 4: I suggest the authors to consider to present data in Figure to improve the visibility of experimental results.

As suggested, we altered the figure 2. We decided to introduce a graphic instead of the chromatograms, please see the lines (374-375).

Point 5: I suggest the authors to give clear recommendations for the industry to decrease the acrylamide concentration of foods and for consumer show to choose food to decrease the intake of acrylamide, respectively.

Although, we are in complete agreement with you regarding the needs to decrease the acrylamide concentration of food, we strongly believe that further in vivo and in vitro studies are needed to be performed in order to re-evaluate the Maximum Residue Limit of acrylamide in food products.

Regarding the recommendations from the very beginning (daily consumption per capita) these information were given in the Results and discussions section  (lines:184, 260, 282-286).

We believe that the quality of the manuscript was much improved based on your recommendations. We trust that you will find our manuscript to meet the requirements for publication in the journal.

Round 2

Reviewer 3 Report

Manuscript foods-1190286 has an interesting topic.  Acrylamide formation is a typical food and food processing related problem. Authors have revised the manuscript thoroughly according to reviwers' comments and suggestions. Manuscript has been well structured and it contains interesting and valuable results. Amendments, reephrasing, additional information, more detailed discussion of experimental results, higher visibility of results made the manuscript more clear and complete. The overall scientific quality of manuscript improved significantly. I accept all answers and modifications made by the authors and recommend manuscript foods-1190286 for publishing.